# Supply-Side Barriers to the Use of Public Healthcare Facilities for Childhood Illness Care in Rural Zambia: A Cross-Sectional Study Linking Data from a Healthcare Facility Census to a Household Survey

**DOI:** 10.3390/ijerph18105409

**Published:** 2021-05-19

**Authors:** Keiji Mochida, Daisuke Nonaka, Jason Wamulume, Jun Kobayashi

**Affiliations:** 1Graduate School of Health Sciences, University of the Ryukyus, 207 Uehara, Nishihara-cho, Nakagami-gun, Okinawa 903-0125, Japan; laodaisuke@hotmail.co.jp (D.N.); junkobalao@gmail.com (J.K.); 2TA Networking Corp., 2-7 Nanpeidai-cho, Shibuya-ku, Tokyo 150-0036, Japan; 3Department of Physical Planning and Medical Technologies, Ministry of Health, Ndeke House, Haile Selassie Avenue, Lusaka P.O. Box 30205, Zambia; jwamulume@gmail.com

**Keywords:** physical access, human resources, equipment, fever, diarrhea, under-five children, Zambia

## Abstract

Child mortality due to malaria and diarrhea can be reduced if proper treatment is received timely at healthcare facilities, but various factors hinder this. The present study assessed the associations between the use of public healthcare facilities among febrile/diarrheal children in rural Zambia and supply-side factors (i.e., the distance from the village to the nearest facility and the availability of essential human resources and medical equipment at the facility). Data from the Demographic and Health Survey 2018 and the Health Facility Census 2017 were linked. Generalized linear mixed models were used to assess the associations, controlling for clustering and other variables. The median distances to the nearest facility were 4.5 km among 854 febrile children and 4.6 km among 813 diarrheal children. Children who were over 10 km away from the facility were significantly less likely to use it, compared to those within 5 km (fever group: odds ratio (OR) = 0.36, 95% confidence interval (CI) = 0.20–0.66; diarrhea group: OR = 0.30, 95% CI = 0.18–0.51). The availability of human resources and equipment was, however, not significantly associated with facility use. Poor geographic access could be a critical barrier to facility use among children in rural Zambia.

## 1. Introduction

Though child mortality has considerably decreased globally in recent years, the mortality rate is still high in sub-Saharan Africa. It was estimated that the under-five mortality rate per 1000 live births was 76 in the region in 2019 compared to the global average of 38 [1]. Malaria and diarrhea are the major causes of child mortality in many countries in the region, where 257,000 and 235,000 children under five died due to these illnesses in 2017, respectively [2].

The mortality from these illnesses can be reduced if patients use a healthcare facility and receive proper treatment in a timely manner, but various factors hinder the use of healthcare facilities. In earlier studies, barriers to health service use in low-income countries in Asia have been categorized into supply- and demand-side barriers [3]. According to this framework, a systematic review, which investigated the factors influencing healthcare-seeking for childhood pneumonia, diarrheal diseases, and malaria in low- and middle-income countries, summarized the geography and cost of healthcare as supply-side barriers and the severity of illness, socioeconomic status, and sex of the child as demand-side barriers [4]. In most settings, supply- and demand-side information are collected separately by facility-based and household surveys, so it is impossible to comprehensively understand the access or barriers to healthcare without merging the data from facility-based and household surveys. A systematic review found 59 articles/conference presentations that merged the data of the supply- and demand-side factors of maternal and child healthcare [5]. However, to the best of our knowledge, few studies have investigated the factors of both sides at the country level in sub-Saharan Africa.

Zambia is one of few sub-Saharan African countries where both supply- and demand-side information is available on a national scale. Similar to other countries in the region, the under-five mortality rate in Zambia remains high, estimated to be 62 for every 1000 live births in 2019 [1]; malaria and diarrhea are the major causes of death, accounting for, 8.2% and 7.8% of the deaths among children under age five in 2017, respectively [2]. In Zambia, these two diseases occupied one fifth of total DALYs among children under age five in 2019 [6], and consequently cause not only health but also social/economic damages. For example, total annual economic impact of malaria in children under age five was estimated $141.5 million, with $114.6 million attributed to productivity losses and $11.7 million in direct costs for the healthcare [7].

Healthcare delivery in Zambia is heavily dependent on public healthcare facilities, including ones managed by mission organizations as they account for 83.2% of total healthcare facilities [8] and the user fee is free for the children under age five there. To manage childhood illness, the Ministry of Health (MOH) set three focus areas: (i) upgrading skills of health workers, (ii) strengthening the health system to deliver the services, and (iii) promoting family and community practices, and care at the community level [9]. As efforts to promote the service delivery, the number of public healthcare facilities dramatically increased from 1340 in 2005 to 2479 in 2017, and numbers of health personnel and basic medical equipment also increased in the same period [8]. However, healthcare services have not been optimally used, especially among children in rural Zambia. The Zambia Demographic and Health Survey (DHS) 2018 revealed that among children who had diarrhea, only 60.9% sought advice or treatment in rural areas, compared to 73.8% in urban areas [10]. 

Therefore, the present study aimed to assess the associations between the use of public healthcare facilities for an episode of fever or diarrhea and supply-side factors in rural Zambia, adjusting for demand-side factors. The supply-side factors for the present study were distance to the healthcare facility and the availability of healthcare personnel and medical equipment. We hypothesized that long distance to the facility and poor availability of the personnel and the equipment decrease utilization of the public healthcare facility. In the present study, children with a fever episode were considered to be potentially suffering from malaria.

## 2. Materials and Methods

### 2.1. Source of Data

This cross-sectional study merged data obtained from a nationwide facility-based survey (i.e., the Zambia Health Facility Census (HFC) 2017) and a household survey (i.e., the Zambia DHS 2018). The Zambia HFC 2017 was carried out by the MOH in financial and technical cooperation with the Japan International Cooperation Agency (JICA) and targeted all public healthcare facilities, including those managed by mission organizations and ministries other than the MOH. In the Zambia HFC 2017, data on infrastructure, human resources, and medical equipment at 2479 facilities were collected through physical enumeration from August 2017 to February 2018 [8]. The Zambia DHS 2018 was a nationally representative household survey conducted by the Zambia Statistics Agency as part of a global DHS program. In the Zambia DHS 2018, two-stage probability proportionate to size sampling drew 12,831 households, and data on the population, health, and nutrition were collected through face-to-face interviews using standardized questionnaires from July 2018 to January 2019 [10].

### 2.2. Participants

A flow diagram of the study participants is presented in Figure 1. We targeted children under the age of five in rural areas (*n* = 6646). The data on age and residential area (i.e., rural/urban) were obtained from the Zambia DHS 2018. We included children who had an episode of fever or diarrhea in the two weeks preceding the Zambia DHS 2018 (*n* = 1151 in the fever group and *n* = 1019 in the diarrhea group).

First, we excluded children without Global Positioning System (GPS) data (*n* = 28 in the fever group and *n* = 21 in the diarrhea group). Second, if multiple children in a household had the same illness, we randomly selected only one child per household (*n* = 118 in the fever group and *n* = 61 in the diarrhea group) in order to simplify the data structure. Third, we excluded children whose closest healthcare facility was neither a health post nor a rural health center (*n* = 70 in the fever group and *n* = 60 in the diarrhea group). Fourth, we excluded children who used a healthcare facility other than the closest health post/rural health center for the illness (*n* = 81 in the fever group and *n* = 64 in the diarrhea group) in order to control the possible influence of supply-side factors, such as type of healthcare facility, other than the predictor variables of interest in the present study. Finally, because the data source for the present study (i.e., the Zambia HFC 2017) contained data on only government and missional healthcare facilities, we excluded children who used private medical and other sectors. Overall, the data of 854 children in the fever group and 813 children in the diarrhea group were analyzed.

### 2.3. Variables and Measurements

The outcome of the present study was the use of a public healthcare facility for an episode of fever or diarrhea among children, which was measured using the Zambia DHS 2018. In the survey, the interviewers asked the caregivers, “Has your child been ill with a fever or diarrhea at any time in the past two weeks?” Furthermore, if the child had either of the symptoms, the survey inquired about the type of healthcare facility or care that was used. Children with fever or diarrhea episodes who used a government health center, government health post, or missional hospital/clinic were categorized as “use public healthcare facility”, while those that did not use any healthcare facilities and those that used a traditional practitioner instead of a healthcare facility were categorized as “not use public healthcare facility”. The classifications of facility type were different between the Zambia HFC 2017 and the Zambia DHS 2018. The government health centers in the Zambia DHS 2018 included both the rural health center and the urban health center in the Zambia HFC 2017. 

The predictor variables of interest in the present study were the distance to the nearest public healthcare facility and the availability of essential human resources and medical equipment at the facility. The geographic coordinates of the clusters from the Zambia DHS 2018 and of healthcare facilities from the Zambia HFC 2017 were used to identify the closest public healthcare facility from each cluster and to measure the distance between facility and cluster. The DHS clusters were villages in rural areas and city blocks in urban areas. We used a straight-line distance between healthcare facilities and clusters because this method has been widely used to measure proximity to a healthcare facility in sub-Saharan Africa [11,12,13]. Although there are several ways to measure geological proximity (i.e., travel distance and travel time), the results obtained by these methods were similar to those obtained in a previous study in Ghana [14]. The straight-line distances in meters were calculated using ArcGIS Pro 2.3.2 version 10.5 (ESRI Inc., Redlands, CA, USA) and classified into three categories according to the thresholds widely used by the MOH [15]: (i) <5 km; (ii) 5–10 km; and (iii) >10 km, as the relationship between the use of the healthcare facility and the distance to the facility was not a linear relationship. 

The availability of essential human resources at the healthcare facility was measured using data from the Zambia HFC 2017. We evaluated available cadres as the ordinal variable according to the standards set by the MOH [8]: (i) Qualified health personnel unavailable; (ii) at least one qualified health worker available, but all four cadres for a standard health post (community health assistant, nurse, midwife, and environmental health personnel) or for a standard rural health center (nurse, midwife, environmental health personnel, and clinical officer) unavailable; (iii) all four cadres for the standard health post available; (iv) all four cadres for the standard rural health center available. 

The availability of essential medical equipment at the healthcare facility was also measured using data from the Zambia HFC 2017 as the ordinal variable: (i) Neither a microscope nor a hemoglobin meter available, (ii) only a microscope available, (iii) only a hemoglobin meter available, and (iv) both of them available. We chose the microscope and the hemoglobin meter, because the MOH requires that all health posts and rural health centers have these two types of equipment [8]. The ages of the studied children were also categorized in line with the standard of DHS reports.

### 2.4. Statistical Analyses

First, descriptive analysis was conducted to determine the characteristics of the studied children and healthcare facilities, assessing their frequency and percentage. Pearson’s chi-square test was used to test for a significant association between the use of a public healthcare facility and supply- and demand-side factors. We used generalized linear mixed models with a binomial distribution and a logit link to assess the association between outcomes and predictors. To account for the clustering of observations, we included the DHS cluster as a random effect in the model. All analyses were performed using SAS University Edition (SAS Institute, Inc., Cary, NC, USA). Additionally, the distance to the nearest public healthcare facility was converted to vigintiles, and the proportions of using the facility in each vigintile were visually inspected.

## 3. Results

### 3.1. Characteristics of the Study Children: Demand-Side Factors

The median age of the study children was 23 months in the fever group and 18 months in the diarrhea group (Table 1). Approximately half of the children were male (48.4% in the fever group and 49.0% in the diarrhea group). The most common educational attainment of their mothers was primary level (64.4% in the fever group and 61.6% in the diarrhea group). Most of the children belonged to the poorest (47.5% in the fever group and 46.4% in the diarrhea group) or the second-poorest (30.4% in the fever group and 30.8% in the diarrhea group) households.

Over half of the children sought advice or treatment at a government health center (54.6% in the fever group and 51.4% in the diarrhea group). Approximately one-fourth of the children did not receive any treatment or advice (23.9% in the fever group and 28.7% in the diarrhea group), and a few received advice or treatment from a traditional practitioner.

### 3.2. Characteristics of Public Healthcare Facilities: Supply-Side Factors

We identified 245 public healthcare facilities as those closest to households with children with fever and 253 facilities as those closest to households with children with diarrhea (Table 2). The median distance between the village and the closest public healthcare facility was 4.5 km for the fever group and 4.6 km for the diarrhea group. Most of the healthcare facilities were categorized as “at least one qualified health worker available” (77.1% for both fever and diarrhea groups). A few healthcare facilities did not have any qualified health personnel (4.1% for the fever group and 3.6% for the diarrhea group). More than half of the healthcare facilities had neither a microscope nor a hemoglobin meter (60.8% for the fever group and 58.5% for the diarrhea group), and only few had both (11.8% for the fever group and 12.3% for the diarrhea group).

### 3.3. Bivariate Analyses for Using Public Healthcare Facilities

Bivariate analysis showed that, among the supply-side characteristics, only distance to the closest public healthcare facility was significantly associated with using the facility for advice or treatment of both fever and diarrhea (Table 3). The highest proportions of using public healthcare facilities were observed among children whose closest public healthcare facility was within 5 km: 80.2% for the fever group and 76.7% for the diarrhea group. The availability of health workers and equipment at the nearest healthcare facility was not associated with using the facility. Among the demand-side factors, for the fever group, the mother’s education and the household’s wealth quintile were significantly associated with using a public healthcare facility. For the diarrhea group, the age of the child was significantly associated with using a public healthcare facility.

A visual inspection of the distance vigintiles showed that the relationship between the use of a healthcare facility and the distance to the facility was not a simple (i.e., non-linear) relationship (Figure 2 and Figure 3). Expectedly, the lowest proportion of healthcare facility use was seen in the last and second to last vigintiles for both the fever and diarrhea groups. Unexpectedly, the highest proportion of healthcare facility use was seen in the 12th vigintile for the fever group and the seventh vigintile for the diarrhea group. 

### 3.4. Multivariate Analyses of the Association between Using a Public Healthcare Facility and Supply-Side Factors

Even after controlling for demand-side factors, such as individual and household characteristics, children in the fever group whose closest public healthcare facility was >10 km from their villages were significantly less likely to use the facility, compared with those whose closest facility was within 5 km (odds ratio (OR) = 0.36; 95% confidence interval (CI) = 0.20 to 0.66) (Table 4). The same significant association was observed in the diarrhea group (OR = 0.30; 95% CI = 0.18 to 0.51). The availability of human resources and equipment was not significantly associated with the outcomes. 

## 4. Discussion

The main finding of the present study was the fact that, although healthcare facility use decreased with increased distance, there was no significant difference in facility use between children living within 5 km from the nearest healthcare facility and those living between 5 and 10 km away. However, there was a significant difference between children living within 5 km from the nearest healthcare facility and those living over 10 km away. This finding suggests that distance has a significant impact on healthcare facility use among children whose villages are located over 10 km away from the nearest healthcare facility.

The main finding confirmed the importance of distance in the use of primary healthcare services in rural Zambia. A cross-sectional study in rural Zambia that used data from the Zambia DHS 2007 showed that the odds of healthcare facility delivery decreased by 29% as the straight-line distance from the village to the closest healthcare facility doubled [12]. Another cross-sectional study showed that the odds of healthcare facility delivery decreased by 65% in rural Malawi and 27% in rural Zambia for every 10 km increase in the straight-line distance from the DHS cluster to the nearest healthcare facility [13]. We also confirmed a negative impact of distance on the use of healthcare facilities for fever and diarrhea care among children from a nationally representative sample after controlling for potential confounders.

Although the main finding elucidates the significant negative impact of a distance of over 10 km to a healthcare facility on treatment seeking, this does not mean that there is no negative impact of a distance of less than 10 km. As shown in Figure 2 and Figure 3, there were large drops in the proportion of using the nearest public healthcare facility among children who live approximately 4–5 km from the nearest healthcare facility. This result is consistent with those of other observational studies and the governments’ norms in sub-Saharan Africa. A study in Kenya showed that the proportion of using healthcare facilities for pediatric fever management decreased among households 5–6 km away from the facilities [16]. The governments of Zambia, as well as other countries in sub-Saharan Africa, aim to ensure that the rural population have access to a healthcare facility within 5 km of where they live [17]. The present study suggests that even a distance of 5 km to a healthcare facility would be a critical barrier for some households (e.g., those who do not own any transport or cannot afford transportation services [18]).

In the present study, the higher proportions of using the public healthcare facility were seen among some vigintiles farther than 4–5 km where the proportion largely dropped. The cross-sectional study in Kenya also showed more frequency of pediatric malaria care use among some of those living farther away from the points where major reduction of the use occurs [16]. Other geographic barriers of the road condition (e.g., a tarmac road and an unpaved road) [19] as well as the ownership of transport and the availability of transportation services may affect the use of public healthcare facility. As the number of households owning any transport was so small, there was not sufficient data to detect to find the relationship with the use of the healthcare facility; and there was no information about transportation services in the villages and road condition to the facility in the present study. Future studies could consider these additional geographic accessibilities on the healthcare facility use.

The efforts to address demand-side challenges and intervention at the community level are never uncontradicted, but the poor geographic access to the healthcare facility could also be one of critical barriers to healthcare at the facility. It is important that the caregivers recognize the diseases for seeking proper treatment in time. A nationally representative household survey, however, showed only 69.3% of women aged 15 to 49 years in rural areas recognized fever as a symptom of malaria [20]. A systematic review found that the care for childhood illness in the community through a community health worker (CHW), who links the community and the healthcare facility, has a large impact on the child survival in sub-Saharan Africa [21]. The percentage of children with diarrhea who received care from CHW was, however, only 2.3% in Zambia [10]. Although more efforts are needed to increase awareness in families and expand care at the community level, the present study showed the importance of geological proximity to the healthcare facility for increasing the use of the facility.

A reason for the insignificant association between the availability of essential health workers and the use of public healthcare facilities in the present study could be that the caregivers believed all health workers in the public healthcare facility to be qualified. A study in rural Uganda showed that caregivers took febrile children to government facilities because they believed that the health providers were better qualified and experienced compared to health providers at drug shops/private clinics [22]. We evaluated the available cadres of health workers, but caregivers might not attach a value to the cadres. 

A possible reason why essential equipment at public healthcare facilities was not significantly associated with using the facility is that caregivers made their decision based on the availability of specific equipment and medicines used for fever/diarrhea care. For example, a study in Zambia in 1997 showed that the major reason caregivers went to a healthcare facility was to obtain medicines when their children had fevers or convulsions. However, if they knew that the facility was out of medicines, they would probably decide not to go there [23]. Although qualitative analysis in the same country indicated the influence of the availability of medicines at a healthcare facility on the use of the facility for malaria care, we had to use the availability of microscopes and hemoglobin meters as proxy variables to measure the general situation of medical equipment. The reason was that there was no information about the availability of rapid diagnostic tests for malaria and first-line antimalarial medicines set by the national diagnosis and treatment guidelines [24]. 

This study has several limitations. First, the distance to the closest public healthcare facility is likely to contain random measurement errors. To protect participant confidentiality, the geographic coordinates of rural DHS clusters were randomly displaced by up to 5 km, with an additional 1% of clusters being displaced by up to 10 km [25]. Despite such random errors, geographical data from the DHS have been widely used in past studies [13,14,16] because it is the only available resource in Zambia and other sub-Sharan African countries.

Second, we assumed the closest healthcare facility from a cluster as a primary source of healthcare and evaluated supply-side factors at the facility; however, children could bypass a facility with poor healthcare quality. A study conducted in Southern Province, Zambia in 2016 compared the healthcare facilities reportedly visited for fever, diarrhea, or cough care and those identified by the straight-line distance as being closest. Most of the children (89% in rural areas) taken for care were linked to their specific reported source of care using the nearest healthcare facility by the straight-line distance [26]. This low proportion of bypassing behavior among children with the same symptom(s), as in the present study, supports the validity of our assumption.

Third, we did not include variables possibly associated with using healthcare facilities. For example, a systematic review found that the cost of healthcare, which is a supply-side factor, is related to using healthcare facilities for child illness, but we could not assess this variable due to a lack of suitable data. Although primary healthcare services are free at public healthcare facilities in Zambia, in a recent study conducted in three provinces, 6.2% of patients reported incurring informal payments, but the incidence was low in rural areas and among maternal and child health service users [27]. This informal cost can be a barrier to using public healthcare facilities in Zambia in general, but may have less of an influence on the association between geological proximity to a public healthcare facility and the use of the facility for child fever and diarrhea.

Despite these limitations, the findings of the present study confirm the importance of implementing the Government of Zambia’s policy to improve geographic access to the healthcare facility: increasing the proportion of rural households living within 5 km of their nearest healthcare facility [17]. Nearly half of the present study’s participants live over 5 km away from public healthcare facilities. As private healthcare facilities are concentrated in large cities, and villages farthest from public healthcare facilities have the greatest needs, the government needs to improve proximity to public healthcare facilities in rural areas to promote childhood illness care at these facilities.

## 5. Conclusions

Poor geographic access could be a critical barrier to obtaining fever and diarrhea care at a healthcare facility among children in rural Zambia, and the negative impact is substantial if the facility is located 10 km or more away from villages. The availability of essential human resources and medical equipment is, however, unlikely to have a significant impact on whether children receive care at the facility. The findings of the present study support Zambia’s national strategy to expand access to public healthcare facilities and emphasize that the villages farthest from the facilities should be prioritized among others. Along with family practices and the care at the community level, more efforts to improve the proximity to healthcare facilities are required to promote childhood illness care at the facilities. Further study should consider the influence of other geographic accessibilities (e.g., the road condition and the availability of transportation services) on healthcare facility use.

## Figures and Tables

**Figure 1 ijerph-18-05409-f001:**
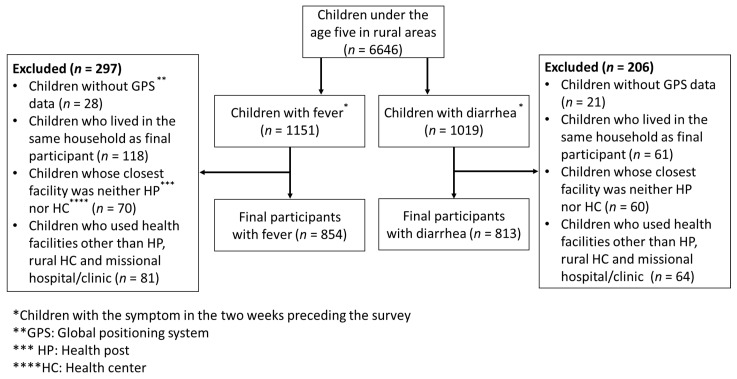
Study participants.

**Figure 2 ijerph-18-05409-f002:**
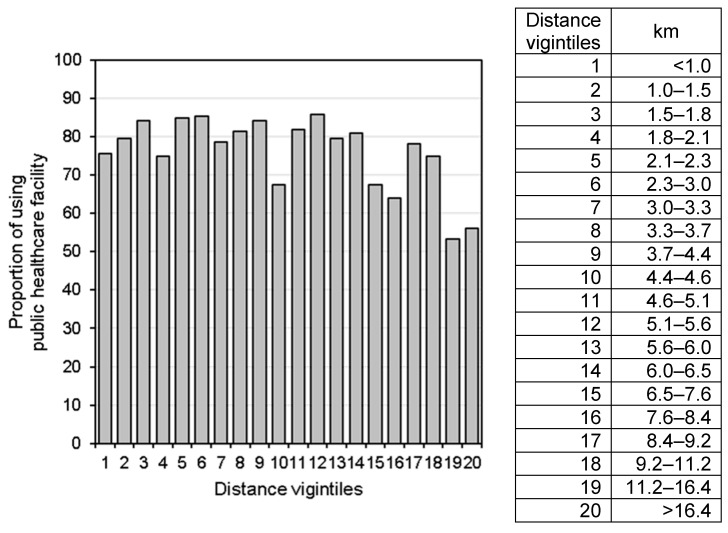
Proportions of using a public healthcare facility for fever care by vigintiles of the distance to the nearest healthcare facility.

**Figure 3 ijerph-18-05409-f003:**
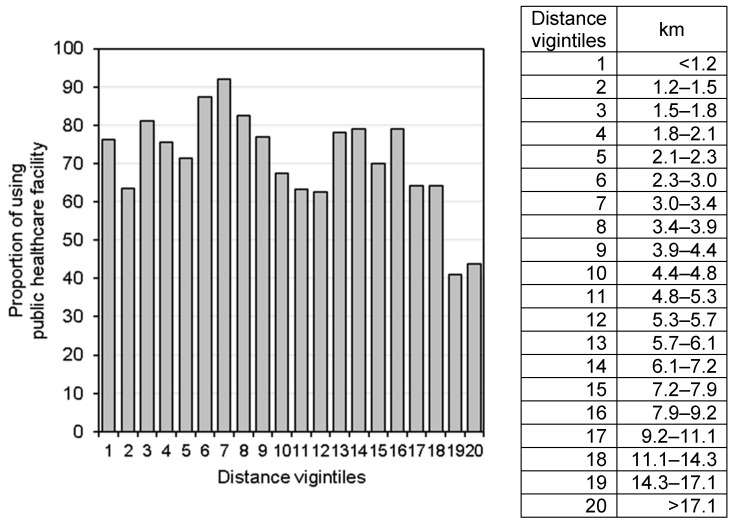
Proportions of using a public healthcare facility for diarrhea care by vigintiles of the distance to the nearest healthcare facility.

**Table 1 ijerph-18-05409-t001:** Characteristics of the studied children: Demand-side factors.

Characteristic	Fever (*n* = 854)		Diarrhea (*n* = 813)	
	*n*	%	*n*	%
Age in months, median (interquartile range)	23	12–35	18	11–28
Sex				
Male	413	48.4	398	49.0
Female	441	51.6	415	51.0
Mother’s educational level				
No formal education	118	13.8	112	13.8
Primary	550	64.4	501	61.6
Secondary	181	21.2	198	24.4
Higher	5	0.6	2	0.2
Household’s wealth quintile				
Lowest	406	47.5	377	46.4
Second	260	30.4	250	30.8
Middle	136	15.9	136	16.7
Fourth	35	4.1	39	4.8
Highest	17	2.0	11	1.4
Source of advice or treatment ^1^				
Government health center	466	54.6	418	51.4
Government health post	159	18.6	145	17.8
Mission hospital/clinic	26	3.0	17	2.1
Traditional practitioner	2	0.2	4	0.5
No treatment	204	23.9	233	28.7

^1^ Since multiple answers were allowed, the numbers do not add up to the total.

**Table 2 ijerph-18-05409-t002:** Characteristics of the public healthcare facilities: Supply-side factors.

Characteristic	Fever (*n* = 245)		Diarrhea (*n* = 253)	
	*n*	%	*n*	%
Distance between cluster and health facility (km), median (interquartile range)	4.5	2.3–7.6	4.6	2.3–7.6
Deployment of health workers				
Qualified health personnel unavailable	10	4.1	9	3.6
At least one qualified health worker available	189	77.1	195	77.1
All four cadres for the standard health post ^1^ available	6	2.4	7	2.8
All four cadres for the standard rural health center ^2^ available	40	16.3	42	16.6
Allocation of equipment				
Neither	149	60.8	148	58.5
Only a microscope	15	6.1	17	6.7
Only a hemoglobin meter	52	21.2	57	22.5
Both	29	11.8	31	12.3

Public healthcare facilities include government hospitals, government health centers, government health posts, and mission hospitals/clinics. ^1^ Midwife, nurse, environmental health personnel, and community health assistant. ^2^ Midwife, nurse, environmental health personnel, and clinical officer.

**Table 3 ijerph-18-05409-t003:** Bivariate associations with using public healthcare facilities.

Characteristic	Fever (*n* = 854)					Diarrhea (*n* = 813)				
	Not Used		Used		*p*-Value ^4^	Not Used		Used		*p*-Value ^3^
	*n*	%	*n*	%		*n*	%	*n*	%	
Supply-side factors										
Distance to the nearest healthcare facility										
<5 km	92	19.8	373	80.2	<0.001	100	23.3	329	76.7	<0.001
5–10 km	69	25.0	207	75.0		69	28.8	171	71.3	
>10 km	44	38.9	69	61.1		67	46.5	77	53.5	
Availability of health workers at the nearest healthcare facility										
Qualified health personnel unavailable	8	29.6	19	70.4	0.756	12	46.2	14	53.8	0.270
At least one qualified health worker available	162	23.9	516	76.1		185	28.5	463	71.5	
All four cadres for the standard health post ^1^ available	4	33.3	8	66.7		5	31.3	11	68.8	
All four cadres for the standard rural health center ^2^ available	31	22.6	106	77.4		34	27.6	89	72.4	
Availability of equipment at the nearest healthcare facility										
Neither	135	26.9	366	73.1	0.121	155	31.1	343	68.9	0.270
Only a microscope	14	20.9	53	79.1		18	31.6	39	68.4	
Only a hemoglobin meter	37	19.4	154	80.6		40	24.1	126	75.9	
Both	19	20.0	76	80.0		23	25.0	69	75.0	
Demand-side factors										
Age in months										
<6	15	23.4	49	76.6	0.851	25	42.4	34	57.6	0.017
6–11	29	21.0	109	79.0		46	27.4	122	72.6	
12–23	61	25.4	179	74.6		80	25.5	234	74.5	
24–35	44	22.1	155	77.9		47	29.0	115	71.0	
36–47	30	26.8	82	73.2		28	42.4	38	57.6	
48–59	26	25.7	75	74.3		10	22.7	34	77.3	
Sex										
Male	93	22.5	320	77.5	0.325	123	30.9	275	69.1	0.248
Female	112	25.4	329	74.6		113	27.2	302	72.8	
Mother’s education										
No formal education	30	25.4	88	74.6	0.049	43	38.4	69	61.6	0.063
Primary	143	26.0	407	74.0		138	27.5	363	72.5	
Secondary or higher	32	17.2	154	82.8		55	27.5	145	72.5	
Household’s wealth quintile										
Lowest	108	26.6	298	73.4	0.009	112	29.7	265	70.3	0.950
Second	56	21.5	204	78.5		69	27.6	181	72.4	
Middle	24	17.6	112	82.4		39	28.7	97	71.3	
Fourth	15	42.9	20	57.1		13	33.3	26	66.7	
Highest	2	11.8	15	88.2		3	27.3	8	72.7	

Public healthcare facilities include government hospitals, government health centers, government health posts, and mission hospitals/clinics. ^1^ Midwife, nurse, environmental health personnel, and community health assistant. ^2^ Midwife, nurse, environmental health personnel, and clinical officer. ^3^ Based on Pearson’s chi-square test.

**Table 4 ijerph-18-05409-t004:** Multivariate association with using public healthcare facilities.

Characteristic	Fever (*n* = 854)		Diarrhea (*n* = 813)	
	AOR	95% CI	AOR	95% CI
Supply-side factors				
Distance to the nearest healthcare facility				
<5 km	1.00	Reference	1.00	Reference
5–10 km	0.74	0.46-1.18	0.73	0.47–1.14
>10 km	0.36	0.20–0.66	0.30	0.18–0.51
Availability of health workers at the nearest healthcare facility				
Qualified health personnel unavailable	1.00	Reference	1.00	Reference
At least one qualified health worker available	1.39	0.47–4.12	2.58	0.95–7.03
All four cadres for the standard health post ^1^ available	0.81	0.13–5.09	2.02	0.41–9.85
All four cadres for the standard rural health center ^2^ available	1.29	0.37–4.54	2.96	0.95–9.24
Availability of equipment at the nearest healthcare facility				
Neither	1.00	Reference	1.00	Reference
Only a microscope	1.08	0.45–2.55	0.67	0.31–1.48
Only a hemoglobin meter	1.29	0.75–2.21	1.21	0.73–2.00
Both	1.36	0.62–3.01	1.32	0.67–2.59
Demand-side factors				
Age in months				
<6	1.00	Reference	1.00	Reference
6–11	1.24	0.55–2.76	1.98	0.99–3.97
12–23	0.91	0.44–1.90	2.10	1.10–4.00
24–35	1.13	0.53–2.41	1.95	0.97–3.92
36–47	0.69	0.31–1.54	0.90	0.40–2.02
48–59	0.71	0.31–1.63	2.57	0.98–6.78
Sex				
Male	1.00	Reference	1.00	Reference
Female	0.88	0.62–1.27	1.15	0.81–1.63
Mother’s education				
No formal education	1.00	Reference	1.00	Reference
Primary	0.84	0.49–1.43	1.47	0.89–2.43
Secondary or higher	1.36	0.68–2.71	1.40	0.76–2.59
Household’s wealth quintile				
Lowest	1.00	Reference	1.00	Reference
Second	1.31	0.85–2.01	0.88	0.58–1.34
Middle	1.56	0.88–2.78	0.83	0.49–1.41
Fourth	0.30	0.13–0.73	0.61	0.26–1.44
Highest	2.25	0.42–12.24	0.92	0.20–4.35

Public healthcare facilities include government hospitals, government health centers, government health posts, and mission hospitals/clinics. ^1^ Midwife, nurse, environmental health personnel, and community health assistant. ^2^ Midwife, nurse, environmental health personnel, and clinical officer. AOR, adjusted odds ratio; CI, confidence interval.

## Data Availability

The datasets of the Zambia DHS 2018 are available upon request from the Demographic Health Surveys. The datasets from the Zambia HFC 2017 are available from the Zambian Ministry of Health on reasonable request.

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
