# Peer review of "Supply-Side Barriers to the Use of Public Healthcare Facilities for Childhood Illness Care in Rural Zambia: A Cross-Sectional Study Linking Data from a Healthcare Facility Census to a Household Survey"

_ijerph, 2021, doi:10.3390/ijerph18105409_

Round 1
Reviewer 1 Report
The paper is well-developed and tackles a relevant issue. Overall, there are just some minor points that should be addessed.
As for the introduction, the background of healthcare in Zambia, and therewith the problem pressure should be outlined more in depth, as the situation may not be familiar to each reader. This includes e.g. the overall damage (socio-economic) stemming from the diseases included in the study, and some information about the status quo of the health system with that respect.
While the methods applied are suitable, it would be helpful to have a better justification why the respective methodology is applied. E.g. why did the authors not use the km distance as a metric measure (but bivariate dependent variables)?
The conclusion is too short, it would be rather nice to read a few more concreate measure here that could be undertaken by the government to improve the situation (beyond the finding in the discussion that the government's policy is suitable). Also further research could be delineated here.
Author Response
Thank you so much for having reviewed our manuscript and provided valuable comments. Our responses to your comments were shown in the file attachment.

Reviewer 2 Report
This manuscript is written well, so I recommend to publish this after minor revision.
1) Authors showed that therefore, the present study aimed to assess the associations between the use of pub~~. However, they did not describe hypothesis in Introduction. At least, it will need to add in the manuscript.
2) Figure 1. (Study participants.) is not easy to see for readers. So, could you revise this again?
Author Response

(The authors gave the same response as above.)

Reviewer 3 Report
Dear Authors
I find you manuscript very pertinent and with higher quality, although long distance to health facilities has been identified in many studies as an important barrier to health care in rural areas.
The only thing I would like to suggest is the conclusion section that is very poor and should be improved, because it is just a small paragraph. You should take this opportunity to recommend that efforts should also focus on involving fathers in child health issues, at the same time empowering about issues of child health care. This includes the need to increase the awareness and understanding of child illnesses and the available preventive and curative services among community members and also be a priority in the health system of Zambia government.
Best wishes and stay safe
Author Response

(The authors gave the same response as above.)
